# FED-$\chi^2$: SECURE FEDERATED CORRELATION TEST

## ABSTRACT

In this paper, we propose the first secure federated $\chi^2$-test protocol, FED-$\chi^2$. We recast $\chi^2$-test as a second frequency moment estimation problem and use stable projection to encode the local information in a short vector. As such encodings can be aggregated with summation, secure aggregation can be applied to conceal the individual updates. We formally establish the security guarantee of FED-$\chi^2$ by demonstrating that the joint distribution is hidden in a subspace containing exponentially possible distributions. Our evaluation results show that FED-$\chi^2$ achieves good accuracy with small client-side computation overhead. FED-$\chi^2$ performs comparably to the centralized $\chi^2$-test in several real-world case studies. The code for evaluation is in the supplementary material.

## 1 INTRODUCTION

Correlation test, as the name implies, is the process of examining the correlation between two random variables using observational data. It is a fundamental building block in a wide variety of real-world applications, including feature selection (Zheng et al., 2004), cryptanalysis (Nyberg, 2001), causal graph discovery (Spirtes et al., 2000), empirical finance (Ledoit & Wolf, 2008; Kim & Ji, 2015), medical studies (Kassirer, 1983) and genomics (Wilson et al., 1999; Dudoit et al., 2003). Because the observational data used in correlation tests may contain sensitive information such as genomic information, centralizing the data collection is risky. To address, we resort to a federated setting in which each client maintains its own data and communicates with a centralized server to calculate a function. Note that the communication should contain as little information as feasible. Otherwise, the server may be able to infer sensitive information from the communication transcript.

In the present work, we study a representative correlation test, namely $\chi^2$-test, under the federated setting. There are two straightforward methods for conducting $\chi^2$-test in such a context. First, clients can upload their raw data to the centralized server and delegate the test to it. While this method is effective in terms of communication, it entirely exposes the clients' private information. Second, clients may run secure multiparty computation (MPC) under the server's coordination. Thus, clients can jointly run $\chi^2$-test without disclosing their data to the server. However, general-purpose MPC imposes significant computation and communication overhead, which is typically intolerable in a federated setting with computationally limited clients, *e.g.*, mobile devices.

To address the dilemma, we present a federated protocol optimized for $\chi^2$-test that is *computationally and communicationally efficient* and *discloses limited information to the server*. We begin by recasting $\chi^2$-test as a second frequency moment estimation problem. To approximate the second frequency moment in a federated setting, each client encodes its raw data into a low-dimensional vector via stable random projection (Indyk, 2006; Vempala, 2005; Li, 2008). Such encodings can be aggregated with only summation, allowing clients to leverage secure aggregation (Bonawitz et al., 2017; Bell et al., 2020) to aggregate the encodings and the server to decode them to approximate the second frequency moment. Because secure aggregation conceals each client's individual update within the aggregated global update, the server learns only limited information about the clients' data.

Our evaluation on four synthetic datasets and 16 real-world datasets shows that FED-$\chi^2$ can replace centralized $\chi^2$-test with good accuracy and low computation overhead. Additionally, we analyze FED-$\chi^2$ in three real-world use cases: feature selection, cryptanalysis, and online false discovery rate control. The results show that FED-$\chi^2$ can achieve comparable performance with centralized $\chi^2$-test and can withstand up to 20% of clients dropping out with minor influence on the accuracy. In summary, we make the following contributions:

- We propose FED-$\chi^2$, the first secure federated $\chi^2$-test protocol. FED-$\chi^2$ is computation- and communication-efficient and leaks much less information than trivially deploying secure aggregation.
- FED-$\chi^2$ decomposes $\chi^2$-test into frequency moments estimation that can easily be encoded/decoded using stable projection and secure aggregation techniques. We give formal security proof and utility analysis of FED-$\chi^2$.
- We evaluate FED-$\chi^2$ in real-world use cases, and the findings suggest that FED-$\chi^2$ can substitute centralized $\chi^2$-test with comparable accuracy, and FED-$\chi^2$ can tolerate up to 20% of clients dropout with minor accuracy drop.

## 2 RELATED WORK

Bonawitz et al. (2017) proposed the well-celebrated secure aggregation protocol as a low-cost way to calculate linear functions in a federated setting. It has seen many variants and improvements since then. For instance, Truex et al. (2019) and Xu et al. (2019) employed advanced crypto tools for secure aggregation, such as threshold homomorphic encryption and functional encryption. So et al. (2021) proposed TURBOAGG, which combines secure sharing with erasure codes for better dropout tolerance. To improve communication efficiency, Bell et al. (2020) and Choi et al. (2020) replaced the complete graph in secure aggregation with either a sparse random graph or a low-degree graph.

Secure aggregation is deployed in a variety of applications. Agarwal et al. (2018) added binomial noise to local gradients, resulting in both differential privacy and communication efficiency. Wang et al. (2020) replaced the binomial noise with discrete Gaussian noise, which is shown to exhibit better composability. Kairouz et al. (2021) proved that the sum of discrete Gaussian is close to discrete Gaussian, thus discarding the common random seed assumption from Wang et al. (2020). The above three works all incorporate secure aggregation in their protocols to lower the noise scale required for differential privacy. Chen et al. (2020) added an extra public parameter to each client to force them to train in the same way, allowing for the detection of malicious clients during aggregation. Nevertheless, designing secure federated correlation tests, despite its importance in real-world scenarios, is not explored by existing research in this field.

On the other end of the spectrum, Wang et al. (2021) proved that stable projection is differentially private if the projection matrix is secret. In our protocol, the projection matrix is public information; hence FED-$\chi^2$ does not consider the differential privacy guarantee.

## 3 FEDERATED CORRELATION TEST WITH MINIMAL LEAKAGE

In this section, we elaborate on the design of FED-$\chi^2$, a secure federated protocol for $\chi^2$-test. Sec. 3.1 first formalizes the problem, establishes the notation system, and introduces the threat model. In Sec. 3.2, we recast $\chi^2$-test as a second frequency moment estimation problem in the federated setting, and consequently, we are able to leverage stable projection to encode each client's local information (Sec. 3.3), and then aggregate them using secure aggregation (Sec. 3.4). Sec. 3.5, 3.6, and 3.7 present security proof, utility analysis, communication analysis, and computation analysis of FED-$\chi^2$.

### 3.1 PROBLEM SETUP

We now formulate the problem of the federated correlation test and establish the notation system. We use $[n]$ to denote $\{1, \cdots, n\}$. We denote vectors with bold lower-case letters (*e.g.*, $\mathbf{a}, \mathbf{b}, \mathbf{c}$) and matrices with bold upper-case letters (*e.g.*, $\mathbf{A}, \mathbf{B}, \mathbf{C}$).

We consider a population of $n$ clients $\mathcal{C} = \{c_i\}_{i \in [n]}$. Each client has one share of local data composed of the triplets $\mathcal{D}_i = \{(x, y, v_{xy}^{(i)})\}, x \in \mathcal{X}, y \in \mathcal{Y}, v_{xy}^{(i)} \in \{-M, \cdots, M\}$, where $x$ and $y$ are categories of the contingency table, $v_{xy}^{(i)}$ is the observed counting of the categories $x$ and $y$ in the local contingency table of the $i^{th}$ client, $|\mathcal{X}| = m_x$ and $|\mathcal{Y}| = m_y$ are finite domains, and $M$ is the maximum value $|v_{xy}^{(i)}|$ can be. The global dataset is defined as $\mathcal{D} = \{(x, y, v_{xy}) : v_{xy} = \sum_{i \in [n]} v_{xy}^{(i)}\}$. We focus on federated $\chi^2$-test and the data in contingency table is discrete. For the ease of presentation, we define the marginal statistics $v_x = \sum_{y \in [|\mathcal{Y}|]} v_{xy}, v_y = \sum_{x \in [|\mathcal{X}|]} v_{xy}$, and $v = \sum_{x \in [|\mathcal{X}|], y \in [|\mathcal{Y}|]} v_{xy}$. Besides, we define $\bar{v}_{xy} = \frac{v_x \times v_y}{v}$, denoting the expectation of $v_{xy}$ if $x$ and

$y$ are uncorrelated. We define $m = m_x m_y$ and use an indexing function $\mathbb{I} : [m_x] \times [m_y] \to [m]$ to obtain a uniform indexing given the indexing of each variable. A centralized server $\mathcal{S}$ calculates the statistics for $\chi^2$-test $s_{\chi^2}(\mathcal{D}) = \sum_{x \in [|\mathcal{X}|], y \in [|\mathcal{Y}|]} \frac{(v_{xy} - \bar{v}_{xy})^2}{\bar{v}_{xy}}$ on the global dataset to decide whether $\mathcal{X}$ and $\mathcal{Y}$ are correlated without collecting the raw data from clients.

Overall, using MPC to conduct secure correlation tests in a federated scenario is highly expensive and impractical (Boyle et al., 2015; Damgård et al., 2012). Hence, in the present work, we trade off accuracy for efficiency, as long as the estimation error is small with a high probability. Formally, if FED-$\chi^2$ outputs $\hat{s}_{\chi^2}$, whose corresponding standard centralized $\chi^2$-test output is $s_{\chi^2}$, the following accuracy requirement should be satisfied with small $\epsilon$ and $\delta$.

$$\mathbb{P}[(1 - \epsilon)s_{\chi^2} \le \hat{s}_{\chi^2} \le (1 + \epsilon)s_{\chi^2}] \ge 1 - \delta$$

**Threat Model.** We assume that the centralized server $\mathcal{S}$ is honest but curious. It honestly follows the protocol due to regulatory or reputational pressure but is curious to discover extra private information from clients' legitimate updates for profit or surveillance purposes. As a result, client updates should contain as little sensitive information as feasible. We want to emphasize that, while the server may explore the privacy of clients, the server will honestly follow the protocol due to regulation or reputational pressure. The server won't provide adversarial vectors to the clients.

On the other hand, we assume honest clients. Specifically, we do not consider client-side adversarial attacks (*e.g.*, data poisoning attacks (Bagdasaryan et al., 2020; Bhagoji et al., 2019)). However, we allow a small portion of clients to drop out during the execution. Also, as we mentioned in Sec. 2, we do not consider the differential privacy guarantee; see further clarification in Appendix A.

## 3.2 FROM CORRELATION TEST TO FREQUENCY MOMENTS ESTIMATION

We first recast correlation test to a second frequency moments estimation problem as defined below. Given a set of key-value pairs $\mathcal{S} = \{k_i, v_i\}_{i \in [n]}$, we re-organize it into a histogram $\mathcal{H} = \{k_j, v_j = \sum_{k_i = k_j, i \in [n]} v_i\}$, and estimate the $\alpha^{th}$ frequency moments as $F_\alpha = \sum_j v_j^\alpha$. $\chi^2$-test can thus be recast to a $2^{nd}$ frequency moments estimation problem as follows:

$$s_{\chi^2}(\mathcal{D}) = \sum_{x,y} \frac{(v_{xy} - \bar{v}_{xy})^2}{\bar{v}_{xy}} = \sum_{x,y} (\frac{v_{xy} - \bar{v}_{xy}}{\sqrt{\bar{v}_{xy}}})^2$$

In federated setting, each client $c_i$ holds a local dataset $\mathcal{D}_i = \{(x, y, v_{xy}^{(i)})\}$ and computes a vector $\mathbf{u}_i$, where $\mathbf{u}_i[\mathbb{I}(x, y)] = \frac{v_{xy}^{(i)} - \bar{v}_{xy}/n}{\sqrt{\bar{v}_{xy}}}$ and $\mathbf{u}_i$ has $m$ elements. Thus, the challenge in federated $\chi^2$-test becomes calculating the following equation:

$$s_{\chi^2}(\mathcal{D}) = \sum_{x,y} (\frac{v_{xy} - \bar{v}_{xy}}{\sqrt{\bar{v}_{xy}}})^2 = || \sum_{i \in [n]} \mathbf{u}_i ||_2^2$$

## 3.3 ENCODING BY STABLE PROJECTION & DECODING BY GEOMETRIC MEAN ESTIMATOR

To address the aforementioned challenges and to easily integrate the algorithm into secure aggregation protocols, we use *stable random projection* (Indyk, 2006; Vempala, 2005) and *geometric mean estimator* (Li, 2008) to approximate the data's second frequency moment efficiently. We begin by discussing stable distributions, followed by the encoding and decoding techniques.

**Definition 1** ($\alpha$-stable distribution). *A random variable $X$ follows an $\alpha$-stable distribution $\mathcal{Q}_{\alpha,\beta,F}$ if its characteristic function is as follows.*

$$\phi_X(t) = \exp(-F|t|^p(1 - \sqrt{-1}\beta \, \text{sgn}(t) \tan(\frac{\pi\alpha}{2})))$$

*, where $F$ is the scale to the $\alpha^{th}$ power and $\beta$ is the skewness.*

$\alpha$-stable distribution is named due to its property called $\alpha$-stability. Briefly, the sum of independent $\alpha$-stable variables still follows an $\alpha$-stable distribution with a different scale.

**Definition 2** ($\alpha$-stability). *If random variables $X \sim \mathcal{Q}_{\alpha,\beta,1}, Y \sim \mathcal{Q}_{\alpha,\beta,1}$ and $X$ and $Y$ are independent, then $C_1 X + C_2 Y \sim \mathcal{Q}_{\alpha,\beta,C_1^\alpha + C_2^\alpha}$.*

We borrow the genius idea from Indyk's well-celebrated paper (Indyk, 2006) to encode the frequency moments in the scale parameter of a stable distribution defined in Definition 1. To encode the local dataset $\mathcal{D}_i = \{(x, y, v_{xy}^{(i)})\}$, each client $c_i, i \in [n]$, is given by the server an $\ell \times m$ projection matrix $\mathbf{P}$ whose values are drawn independently from an $\alpha$-stable distribution $\mathcal{Q}_{\alpha,\beta,1}$, where $\ell$ is the encoding

---

**Algorithm 1:** The encoding and decoding scheme for federated frequency moments estimation. Note that the encoding and decoding themselves do not provide any security guarantee.

---

**1 Function** ENCODE ($P$, $u_i$):
**2**      **return** $\mathbf{P} \times \mathbf{u}_i$
**3 Function** GEOMETRICMEANESTIMATOR ($e$):
**4**      $\hat{d}_{(2),gm} \leftarrow \frac{\prod_{k=1}^{\ell} |\mathbf{e}_k|^{2/\ell}}{(\frac{2}{\pi}\Gamma(\frac{2}{\ell})\Gamma(1-\frac{1}{\ell})\sin(\frac{\pi}{\ell}))^{\ell}}$               `// ℓ is the encoding size.`
**5**      **return** $\hat{d}_{(2),gm}$
**6 Function** DECODE ($e$):
**7**      **return** GEOMETRICMEANESTIMATOR ($e$)

---

size and $m = m_x m_y$. The client re-organizes the data into a vector $\mathbf{u}_i$, where $\mathbf{u}_i[\mathbb{I}(x,y)] = v_{xy}^{(i)}$. Then, the client projects $\mathbf{u}_i$ to $\mathbf{e}_i = \mathbf{P} \times \mathbf{u}_i$ as the encoding (line 2 in Alg. 1).

To decode, the server first sums the encodings up $\mathbf{e} = \sum_{i \in [n]} \mathbf{e}_i$ and estimates the scale of the variables in the aggregated encoding with an unbiased geometric mean estimator (Li, 2008) in line 4 of Alg. 1. According to the $\alpha$-stability defined in Definition 2, every element $e_k$ in $\mathbf{e}$, $k \in [\ell]$, follows this stable distribution: $e_k \sim \mathcal{Q}_{2,0,||\sum_{i \in [n]} \mathbf{u}_i||_2^2}$. Thus, the $l_2$ norm can be estimated by calculating the scale of the distribution $\mathcal{Q}_{2,0,||\sum_{i \in [n]} \mathbf{u}_i||_2^2}$ with $\mathbf{e}$ containing $\ell$ elements.

During the above processes, $\mathbf{u}_i$ and $\sum_{i \in [n]} \mathbf{u}_i$ are not leaked and the aggregation among clients is calculated with only summation. Thus, secure aggregation protocols can be naturally applied with quantization, as will be discussed in Sec. 3.4 and 3.5. Furthermore, $\ell = \frac{c}{\epsilon^2} \log(1/\delta)$ suffices to guarantee that the second frequency moment can be approximated with a $1 \pm \epsilon$ factor and a probability no less than $1 - \delta$. We will further analyze the utility of FED-$\chi^2$ in Sec. 3.6.

### 3.4 SECURE FEDERATED CORRELATION TEST

The complete protocol for FED-$\chi^2$ is presented in Alg. 2. Firstly, the marginal statistics $v_x, v_y$ and $v$ are calculated with secure aggregation and broadcasted to all clients (lines 1–6 in Alg. 2). This step can be omitted if the marginal statistics are already known. Then, on the server side, a projection matrix $\mathbf{P}$ is sampled from an $\alpha$-stable distribution $\mathcal{Q}_{2,0,1}^{\ell \times m}$. The projection matrix is broadcasted to all clients (lines 8–10 in Alg. 2). For each client $c_i$, the local data is re-organized into $\mathbf{u}_i$ and projected to $\mathbf{e}_i$ as encoding (lines 11–14 in Alg. 2). Then, the encoding results will be quantized and aggregated with secure aggregation (line 15 in Alg. 2). As we have already known the marginal statistics in the first round, the quantization bound can be set accordingly. Additionally, we can use high precision for quantization, such as 64 bits, since the size of the contingency table is normally moderate rather than enormous. Thus, the precision of the quantized float number is comparable to or even better than that of float64, and hence we disregard the effect of quantization on accuracy. Finally, the server gets the $\chi^2$-test result using the decoding algorithm described in Alg. 1 (line 17 in Alg. 2).

Dropouts in the first round have no effect on the test's accuracy because they can be recovered inside secure aggregation (Bonawitz et al., 2017; Bell et al., 2020). On the other hand, dropouts in the second round will affect the accuracy of the test. Still, because the $\chi^2$ value is typically far from the decision threshold, FED-$\chi^2$ is intrinsically robust to a small portion of clients dropping out (see Section 4 for empirical assessment).

### 3.5 SECURITY ANALYSIS

As discussed in Sec. 2, the secure aggregation protocol is well studied, and many well-celebrated secure aggregation protocols have been proposed (Bonawitz et al., 2017; Truex et al., 2019; Xu et al., 2019; So et al., 2021; Bell et al., 2020; Choi et al., 2020). In this paper, we choose the state-of-the-art secure aggregation protocol by Bell et al. (Bell et al., 2020), which replaces the complete graph with a sparse random graph to enhance communication efficiency. We clarify that FED-$\chi^2$ can incorporate other popular secure aggregation protocols. We now prove the security enforced by Alg. 2 via a standard simulation proof process (Lindell, 2017) on the basis of Theorem 1.

**Theorem 1** (Security). *Let $\Pi$ be an instantiation of Alg. 2 with the secure aggregation protocol in Alg. 4 of Appendix H with cryprographic security parameter $\lambda$. There exists a PPT simulator SIM such that for all clients $\mathcal{C}$, the number of clients $n$, all the marginal distributions $\{v_x\}, \{v_y\}$, and all*

*the encodings $\{e_i\}$, the output of* SIM *is indistinguishable from the view of the real server* $\Pi_{\mathcal{C}}$ *in that execution, i.e.,* $\Pi_{\mathcal{C}} \approx_{\lambda}$ SIM$(\sum e_i, n)$.

Intuitively, Theorem 1 illustrates that no more information about the clients except the averaged updates is revealed to the centralized server. Thus, each client's update is hidden by the rest clients in secure aggregation. We now present the formal proof for Theorem 1.

*Proof for Theorem 1.* To prove Theorem 1, we need the following lemma.

**Lemma 1** (Security of secure aggregation protocol)**.** *Let* SECUREAGG *be the secure aggregation protocol in Alg. 4 of Appendix H instantiated with cryprographic security parameter* $\lambda$*. There exists a probabilistic polynomial-time (PPT) simulator* SIMSA *such that for all clients* $\mathcal{C}$*, the number of clients* $n$*, and all inputs* $\mathcal{X} = \{e_i\}_{i \in [n]}$*, the output of* SIMSA *is perfectly indistinguishable from the view of the real server, i.e.,* SECUREAGG$_{\mathcal{C}} \approx_{\lambda}$ SIMSA$(\sum_{i \in [n]} e_i, n)$.

Lemma 1 is derived from the security analysis of our employed secure aggregation protocol (Theorem 3.6 in Bell et al. (2020)), which establishes that the secure aggregation protocol securely conceals the individual information in the aggregated result. With this lemma, we are able to prove the theorem for federated $\chi^2$-test by presenting a sequence of hybrids that begin with real protocol execution and end with simulated protocol execution. We demonstrate that every two consecutive hybrids are indistinguishable, illustrating that the hybrids are indistinguishable according to transitivity.

HYB$_1$ This is the view of the server in the real protocol execution, REAL$_{\mathcal{C}}$.

HYB$_2$ In this hybrid, we replace the view during the execution of each SECUREAGG$(\{v_x^{(i)}\}_{i \in [n]})$ in line 3 of Alg. 2 with the output of SIMSA$(v_x, n)$ one by one. According to Lemma 1, each replacement does not change the indistinguishability. Hence, HYB$_2$ is indistinguishable from HYB$_1$.

HYB$_3$ Similar to HYB$_2$, we replace the view during the execution of each SECUREAGG$(\{v_y^{(i)}\}_{i \in [n]})$ in line 4 of Alg. 2 with the output of SIMSA$(v_y, n)$ one by one. According to Lemma 1, HYB$_3$ is indistinguishable from HYB$_2$.

HYB$_4$ In this hybrid, we replace the view during the execution of SECUREAGG$(\{e_i\}_{i \in [n]})$ in line 15 of Alg. 2 with the SIMSA$(\sum e_i, n)$. This hybrid is the output of SIM. According to Lemma 1, HYB$_4$ is indistinguishable from HYB$_3$. $\qquad\square$

**Remark: what does Alg. 2 leak?** By Theorem 1, we show that individual updates of clients are perfectly hidden in the aggregated results and FED-$\chi^2$ leaks no more than a linear equation system:

$$\begin{cases} \mathbf{P} \times \mathbf{v} & = \mathbf{e}^T \\ \mathbf{J}_{1,m_y} \times \mathbf{V}^T & = \mathbf{v}_x^T \\ \mathbf{J}_{1,m_x} \times \mathbf{V} & = \mathbf{v}_y^T \end{cases}$$

---

**Algorithm 2:** FED-$\chi^2$: secure federated $\chi^2$-test. SECUREAGG is a remote procedure that receives inputs from the clients and returns the summation to the server. INITSECUREAGG is the corresponding setup protocol deciding the communication graph and other hyper-parameters.

1 **Round** 1: Reveal the marginal statistics
2    INITSECUREAGG $(n)$                        // $n$ is the client number.
3    **for** $x \in [m_x]$ **do** $v_x = $ SECUREAGG $(\{v_x^{(i)}\}_{i \in [n]})$
4    **for** $y \in [m_y]$ **do** $v_y = $ SECUREAGG $(\{v_y^{(i)}\}_{i \in [n]})$
5    **Server**
6      Calculate $v = \sum_x v_x$ and broadcast $v$, $\{v_x\}$ and $\{v_y\}$ to all the clients.
7 **Round** 2: Approximate the statistics
8    **Server**
9      Sample the projection matrix $\mathbf{P}$ from $\mathcal{Q}_{2,0,1}^{\ell \times m}$
10      Broadcast the projection matrices to the clients
11    **Client** $c_i, i \in [n]$
12      Calculate $\bar{v}_{xy} = \frac{v_x v_y}{v}$
13      Prepare $\mathbf{u}_i$ s.t. $\mathbf{u}_i[\mathbb{I}(x,y)] = \frac{v_{xy}^{(i)} - \bar{v}_{xy}/n}{\sqrt{\bar{v}_{xy}}}$
14      Calculate $\mathbf{e}_i = $ ENCODE $(\mathbf{P}, \mathbf{u}_i)$
15    $\mathbf{e} = $ SECUREAGG (QUANTIZE $(\{e_i\}_{i \in [n]})$)
16    **Server**
17      $\hat{s}_{\chi^2} = $ DECODE $(\mathbf{e})$

, where $\mathbf{J}_{1,m_x}$ and $\mathbf{J}_{1,m_y}$ are $1 \times m_x$ and $1 \times m_y$ unit matrices, $\mathbf{V}$ is an $m_x \times m_y$ matrix whose elements are $\{v_{xy}\}$, and $\mathbf{v}$ is a vector flattened by $\mathbf{V}$. To understand this, information leaked by FED-$\chi^2$ includes the estimation $\mathbf{e}$ and marginal statistics $\mathbf{v}_x$ and $\mathbf{v}_y$. The following theorem establishes an important fact: the above equation system has an exponentially large solution space, which effectively conceals the real joint distribution. We thus believe that Alg. 2 practically ensures privacy due to the solution space's vastness.

**Theorem 2.** *Given a projection matrix $\boldsymbol{P} \in \mathbb{Z}_q^{\ell \times m}$, $\{v_x\}$, $\{v_y\}$ and $\boldsymbol{e}$, there are at least $q^{m-\ell-m_x-m_y}$ feasible choices of $\{v_{xy}\}$.*

*Proof sketch for Theorem 2.* As demonstrated above, the given information forms a linear equation with $m_x + m_y + \ell$ equations. Given $m > m_x + m_y + \ell$, the rank of the coefficient matrix is no more than $m_x + m_y + \ell$. Solving the equations with Smith normal form, we know that the solution space is at least $(m - m_x - m_y - \ell)$-dimensional. With the following lemma, we manage to prove Theorem 2.

**Lemma 2.** *There are $q^{r \times c}$ vectors in the subspace of $\mathbb{Z}_q^{r \times c}$.* □

## 3.6 UTILITY ANALYSIS

In this section, we conduct the utility analysis in terms of multiplicative error. We show that the output of FED-$\chi^2$, $\hat{s}_{\chi^2}$, is a fairly accurate approximation (parameterized by $\epsilon$) to the correlation test output $s_{\chi^2}$ in the standard centralized setting with high probability parameterized by $\delta$.

**Theorem 3** (Utility). *Let $\Pi$ be an instantiation of Alg. 2 with secure aggregation protocol in Alg. 4 of Appendix H. $\Pi$ is parameterized with $\ell = \frac{c}{\epsilon^2} \log(1/\delta)$ for some constant $c$. After executing $\Pi_{\mathcal{C}}$ on all clients $\mathcal{C}$, the server yields $\hat{s}_{\chi^2}$, whose distance to the accurate correlation test output $s_{\chi^2}$ is bounded with high probability as follows:*

$$\mathbb{P}[\hat{s}_{\chi^2} < (1-\epsilon)s_{\chi^2} \vee \hat{s}_{\chi^2} > (1+\epsilon)s_{\chi^2}] \leq \delta$$

*Proof sketch for Theorem 3.* First, we introduce the following lemma from Li (2008).

**Lemma 3** (Tail bounds of geometric mean estimator (Li, 2008)). *The right tail bound of geometric mean estimator is:*

$$\mathbb{P}(\hat{s}_{\chi^2} - s_{\chi^2} > \epsilon s_{\chi^2}) \leq \exp(-\ell \frac{\epsilon^2}{G_R})$$

*where $\frac{\epsilon^2}{G_R} = C_1 \log(1+\epsilon) - C_1 \gamma_e (\alpha - 1) - \log(\frac{2}{\pi} \Gamma(\alpha C_1) \Gamma(1 - C_1) \sin(\frac{\pi \alpha C_1}{2}))$, $\alpha = 2$ in our setting, $C_1 = \frac{2}{\pi} \tan^{-1}(\frac{\log(1+\epsilon)}{(2+\alpha^2)\pi/6})$, and $\gamma_e = 0.577215665...$ is the Euler's constant.*

*The left tail bound of the geometric mean estimator is:*

$$\mathbb{P}(\hat{s}_{\chi^2} - s_{\chi^2} < -\epsilon s_{\chi^2}) \leq \exp(-\ell \frac{\epsilon^2}{G_L})$$

*where $\ell > \ell_0$, $\frac{\epsilon^2}{G_L} = -C_2 \log(1-\epsilon) - \log(-\frac{2}{\pi} \Gamma(-\alpha C_2) \Gamma(1 + C_2) \sin(\frac{\pi \alpha C_2}{2})) - \ell_0 C_2 \log(\frac{2}{\pi} \Gamma(\frac{\alpha}{\ell_0}) \Gamma(1 - \frac{\alpha}{\ell_0}) \sin(\frac{\pi}{2} \frac{\alpha}{\ell_0}))$, and $C_2 = \frac{12}{\pi^2} \frac{\epsilon}{(2+\alpha^2)}$.*

With Lemma 3, Taking $c \geq \max(G_R, G_L)$ and $\delta = \exp(-\frac{\ell \epsilon^2}{c})$, we are able to prove $\mathbb{P}[\hat{s}_{\chi^2} < (1-\epsilon)s_{\chi^2} \vee \hat{s}_{\chi^2} > (1+\epsilon)s_{\chi^2}] \leq \delta$, and the above bound holds when $\ell = \frac{c}{\epsilon^2} \log(1/\delta)$. □

## 3.7 COMMUNICATION & COMPUTATION ANALYSIS

In this section, we present the communication and computation cost of Alg. 2.

**Theorem 4** (Communication Cost). *Let $\Pi$ be an instantiation of Alg. 2 with secure aggregation protocol in Alg. 4 of Appendix H, then (1) the client-side communication cost is $\mathcal{O}(\log n + m_x + m_y + \ell)$; (2) the server-side communication cost is $\mathcal{O}(n \log n + n m_x + n m_y + n\ell)$.*

**Theorem 5** (Computation Cost). *Let $\Pi$ be an instantiation of Alg. 2 with secure aggregation protocol in Alg. 4 of Appendix H, then (1) the client-side computation cost is $\mathcal{O}(\log^2 n + (\ell + m_x + m_y) \log n + m\ell)$; (2) the server-side computation cost is $\mathcal{O}(n \log^2 n + n(\ell + m_x + m_y) \log n + \ell)$.*

Note that compared with the original computation cost presented in (Bell et al., 2020), the client-side overhead has an extra $\mathcal{O}(m\ell)$ term. This term is incurred by the encoding overhead. We also give an empirical evaluation on the client-side computation overhead in Sec. 4.1. Please refer to Appendix I for the detailed proof of Theorem 4 and Theorem 5.

# 4 EVALUATION

**Experiment Setup.** To assess FED-$\chi^2$'s accuracy, we simulate it on four synthetic datasets and 16 real-world datasets. We compare the multiplicative errors of FED-$\chi^2$ with that of the standard centralized $\chi^2$-test. The four synthetic datasets are independent, linearly correlated, quadratically correlated, and logistically correlated. As the real-world datasets, we report the details in Appendix J.

Additionally, we evaluate FED-$\chi^2$'s utility in three real-world application scenarios: 1) feature selection, 2) cryptanalysis, and 3) online false discovery rate (FDR) control. For feature selection, we report the model accuracy trained on the selected features. For cryptanalysis, we report the success rate of cracking ciphertexts. For Online FDR control, we report the average false discovery rate. We compare the performance of FED-$\chi^2$ with that of the centralized $\chi^2$-test in each of the three experiments. Unless otherwise specified, experiments are launched on an Ubuntu 18.04 LTS server equipped with 32 AMD Opteron(TM) Processor 6212 and 512GB RAM.

## 4.1 EVALUATION RESULTS

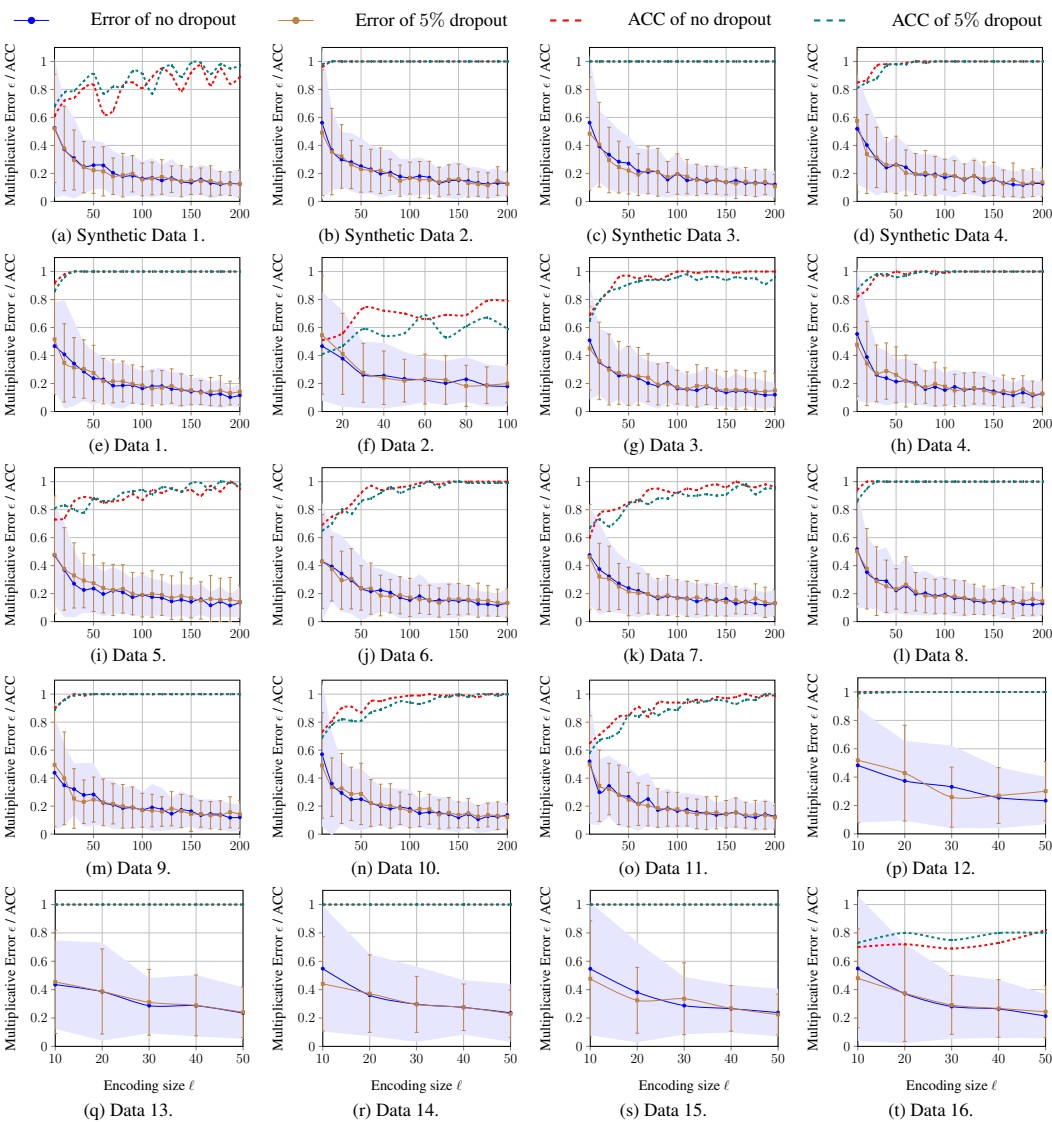

Figure 1: Multiplicative error and accuracy of FED-$\chi^2$ w.r.t. encoding size $\ell$ w/ and w/o dropout.

**Accuracy.** We begin by evaluating the accuracy of FED-$\chi^2$, as illustrated in Fig. 1. Each point represents the mean of 100 independent runs with 100 clients, while the error bars indicate the standard deviation. We choose $m_x = m_y = 20$ in this experiment. Note that the accuracy drop is independent of the number of clients.

From Fig. 1, we observe that the larger the encoding size $\ell$, the smaller the multiplicative error. When $\ell = 50$, the multiplicative error $\epsilon \approx 0.2$. This conforms with Theorem 3, in which the multiplicative error $\epsilon = \sqrt{\frac{c}{\ell} \log(1/\delta)}$ decreases as $\ell$ increases.

We also evaluate the power (Cohen, 2013) of FED-$\chi^2$. We set the p-value threshold as $0.05$, which determines whether or not to reject the null hypothesis. From The dashed lines in Fig. 1, we can tell that the power of FED-$\chi^2$ is high. This conforms with our observation on the multiplicative errors. Specifically, since the $\chi^2$ values are typically far from the decision threshold, a multiplicative error of $0.2$ rarely flips the final decision.

We also present the results when $5\%$ of clients drop out in the second round of FED-$\chi^2$ in Fig. 1. The results show that FED-$\chi^2$ is robust to a small portion of dropouts. In Appendix G, we present the results in terms of $10\%$, $15\%$, and $20\%$ dropout rates.

**Client-side Computation Overhead.** To assess extra computation overhead incurred by FED-$\chi^2$ on the client side, we measure the execution time of the encoding scheme on an Android 10 mobile device equipped with a Snapdragon865 CPU and 12GB RAM. We use PyDroid (Sandeep Nandal, 2020) to run the client-side computation of FED-$\chi^2$ on the Android device.

The results are shown in Fig. 2. Each point represents the average of 100 separate runs, with accompanying error bars. The overhead is generally negligible. For example, for a $500 \times 500$ contingency table, the encoding takes less than 30ms. The overhead grows linearly in relation to $m_x$ ($m_y$) and consequently quadratically in Fig. 2, where $m_x$ equals $m_y$.

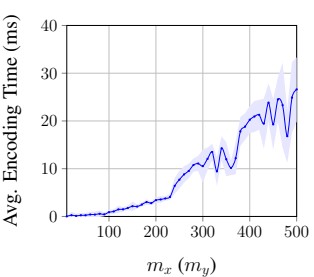

Figure 2: Client-side encoding overhead when $m_x = m_y$.

## 4.2 DOWNSTREAM USE CASE STUDY

**Feature Selection.** Our first case study explores secure federated feature selection using FED-$\chi^2$. The setting is that each client holds data with a large feature space and wants to collaborate with other clients to rule out unimportant features and retain features with top-$k$ highest $\chi^2$ scores. We use Reuters-21578 (Hayes & Weinstein, 1990), a standard text categorization dataset (Yang, 1999; Yang & Pedersen, 1997; Zhang & Yang, 2003), and pick the top-20 most frequent categories using 17,262 training and 4,316 test documents. These documents are distributed randomly to 100 clients, each of whom receives the same number of training documents. After removing all numbers and stop-words, we obtain 167,135 indexing terms. After performing feature selection using FED-$\chi^2$, we select the top 40,000 terms with the highest $\chi^2$ scores. When compared with the centralized $\chi^2$-test, 38,012 (95.03%) of the selected terms are identical, indicating that FED-$\chi^2$ produces highly consistent results with the standard $\chi^2$-test.

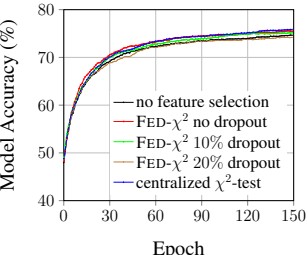

Figure 3: Accuracy of the model trained with features selected by FED-$\chi^2$ and centralized $\chi^2$-test.

We then train logistic regression models using the terms selected by FED-$\chi^2$ and the centralized $\chi^2$-test, respectively. All hyper-parameters are the same. The details of these models are reported in Appendix K. The results in Fig. 3 further demonstrate that FED-$\chi^2$ exhibits comparable performance with the centralized $\chi^2$-test. When $10\%$ and $20\%$ of clients dropout in the second round of FED-$\chi^2$, the accuracy of the trained model using the features selected by FED-$\chi^2$ does not drop much. We also examine performance without feature selection, and as expected, model accuracy is significantly greater after feature selection. Note that the model without feature selection has 2,542,700 more parameters than the model with feature selection. Hence, feature selection effectively improves model accuracy while reducing model size and computational cost.

**Cryptanalysis.** In the second case study, we explore federated cryptanalysis with FED-$\chi^2$. We break Caesar cipher (Luciano & Prichett, 1987), a classic substitution cipher, with FED-$\chi^2$. In a Caesar cipher, each letter in the plaintext is replaced by another letter with some fixed number of positions down the alphabet. For instance, each English letter can be right-shifted by three, converting

the plaintext "good" to the ciphertext "jrrg". There are 26 possible shifts when given 26 English letters. The plaintext can be cracked in a shortcut by performing a correlation test on the ciphertext in relation to normal English text.

In our setting, each client is assumed to possess a segment of the Caesar ciphertext. To collaboratively crack the ciphertext, these clients run 26 $\chi^2$-tests to determine the correlation level between each ciphertext letter and the letters in normal English text. The $\chi^2$-test yielding the highest correlation level elucidates how English letters are encrypted into Caesar ciphertexts.

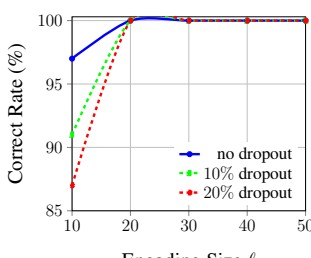

Figure 4: The success rate of cracking Caesar ciphers.

We take Shakespeare's lines as the plaintext and encrypt it into a Caesar ciphertext with a length of 1000 characters. We initiate the cracking process on ten non-overlapping ciphertexts to compute the average success rate (see Fig. 4). In general, the larger the encoding size, the more precise the $\chi^2$-statistics is, and consequently the higher the success rate. Again, according to Theorem 3, the multiplicative error $\epsilon$ decreases as the encoding size increases. In Fig. 4, we also report the success rate when 10% and 20% of clients dropout in Round two of FED-$\chi^2$, respectively. Even if 20% of clients dropout, the success rate can still be 100% as long as the encoding size $\ell$ is larger than 20.

**Online False Discovery Rate Control.** In the third case study, we explore federated online false discovery rate (FDR) control (Foster & Stine, 2008) with FED-$\chi^2$. In an online FDR control problem, a data analyst receives a stream of hypotheses on the database, or equivalently, a stream of $p$-values: $p_1, p_2, \cdots$. At each time $t$, the data analyst should pick a threshold $\alpha_t$ to reject the hypothesis when $p_t < \alpha_t$. The error metric is the false discovery rate, and the objective of online FDR control is to ensure that for any time $t$, the FDR up to time $t$ is smaller than a pre-determined quantity.

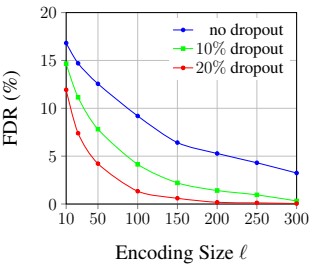

Figure 5: Average FDR w.r.t. $\ell$ for SAFFRON with FED-$\chi^2$.

We use the SAFFRON procedure (Ramdas et al., 2018), the state-of-the-art online FDR control, for multiple hypothesis testing. The $\chi^2$ results and corresponding $p$-values are calculated by FED-$\chi^2$. We present the detailed algorithm of SAFFRON and the hyper-parameters used in our evaluation in Appendix L. The size of the randomly synthesized contingency table is $20 \times 20$. Each time, there are 100 independent hypotheses, with a probability of 0.5 that each hypothesis is either independent or correlated. The time sequence length is 100, and the number of clients is 10. The data are synthesized from a multivariate Gaussian distribution. For the correlated data, the covariance matrix is randomly sampled from a uniform distribution. For the independent data, the covariance matrix is diagonal, and its entries are randomly sampled from a uniform distribution.

At time $t$, we use FED-$\chi^2$ to calculate the $p$-values $p_t$ of all the hypotheses, and then use the SAFFRON procedure to estimate the reject threshold $\alpha_t$ using $p_t$. The relationship between the average FDR and encoding size $\ell$ is shown in Fig. 5. We observe that the variance of independent runs is very small, so we omit the error bars. The results indicate that by increasing the encoding size $\ell$, FED-$\chi^2$ can achieve a low FDR of less than 5.0%. We also observe that dropouts improve the performance of FED-$\chi^2$ in this case study. The reason for this is because dropouts reduce the estimated $\chi^2$ value, which increases the probability of accepting the null hypothesis in FED-$\chi^2$. As a larger portion of queries follows the null hypothesis in online FDR control, the accuracy also increases. The results further demonstrate that FED-$\chi^2$ can be employed in practice to facilitate online FDR control using a secure federated correlation test.

## 5 CONCLUSION

This paper takes an important step towards designing non-linear secure aggregation protocols in the federated setting. Specifically, we propose a universal secure protocol to evaluate frequency moments in the federated setting. We focus on an important application of the protocol: $\chi^2$-test. We give formal security proof and utility analysis on our proposed secure federated learning $\chi^2$-test protocol FED-$\chi^2$ and validate them with empirical evaluations and case studies.

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

# APPENDIX

## A    CLARIFICATION ON PRIVACY

As indicated in Sec. 2, the projection matrix is public information, and hence FED-$\chi^2$ does not take the differential privacy guarantee into account. We would want to provide more information in order to eliminate any potential misunderstandings.

To begin, we would want to emphasize that "privacy" in our paper refers to MPC-style privacy, not DP-style privacy. In general, MPC-style privacy is orthogonal to DP-style privacy: in MPC, privacy is obtained against a semi-honest server in such a way that the server cannot witness individual client's updates but only an aggregate of them, e.g. SecAgg (Bonawitz et al., 2017). In DP, privacy is accomplished by including random noise in each client's update, such that the distribution of the output result does not reveal the clients' private information and the server cannot infer the clients' identification from the output result.

Second, we would like to emphasize that our work proposes a novel secure aggregation scheme particularly for the $\chi^2$-test. Existing standard secure aggregation schemes are inapplicable to the $\chi^2$-test, which will reveal much more information than FED-$\chi^2$, as we have clarified in Sec. 1. Again, this work requires guaranteeing MPC-style privacy, not DP.

Third, to quantify MPC-style privacy, we prove in Theorem 2 that the clients' updates in FED-$\chi^2$ are hidden inside a space with exponential size. This is weaker than hiding users' updates in the whole space, but still gives meaningful privacy guarantees (consider attempting to guess the output of an exponential-sided dice, which is practically infeasible).

Finally, while DP is orthogonal to this research, we would want to emphasize that our protocol can achieve DP by introducing calibrated discrete Gaussian noise to the users' local updates.

## B    COMPARISON WITH POOLING $\chi^2$-TEST

We also compare the performance of FED-$\chi^2$ with pooling $\chi^2$-test. That is, the clients compute the $\chi^2$-test with their local observations and then they aggregate their test results by pooling. The result of the pooling $\chi^2$-test is determined by the majority of the clients' local results. Fig. 6 shows the result of pooling $\chi^2$-test on the real-world datasets. The details of the datasets are presented in Appendix J. We observe that pooling $\chi^2$-test cannot give meaningful results and it tends to give judgement that the data is independent since that the numbers of the local observations are not sufficient to make correct judgement. The results further demonstrate the effectiveness and the necessity of FED-$\chi^2$.

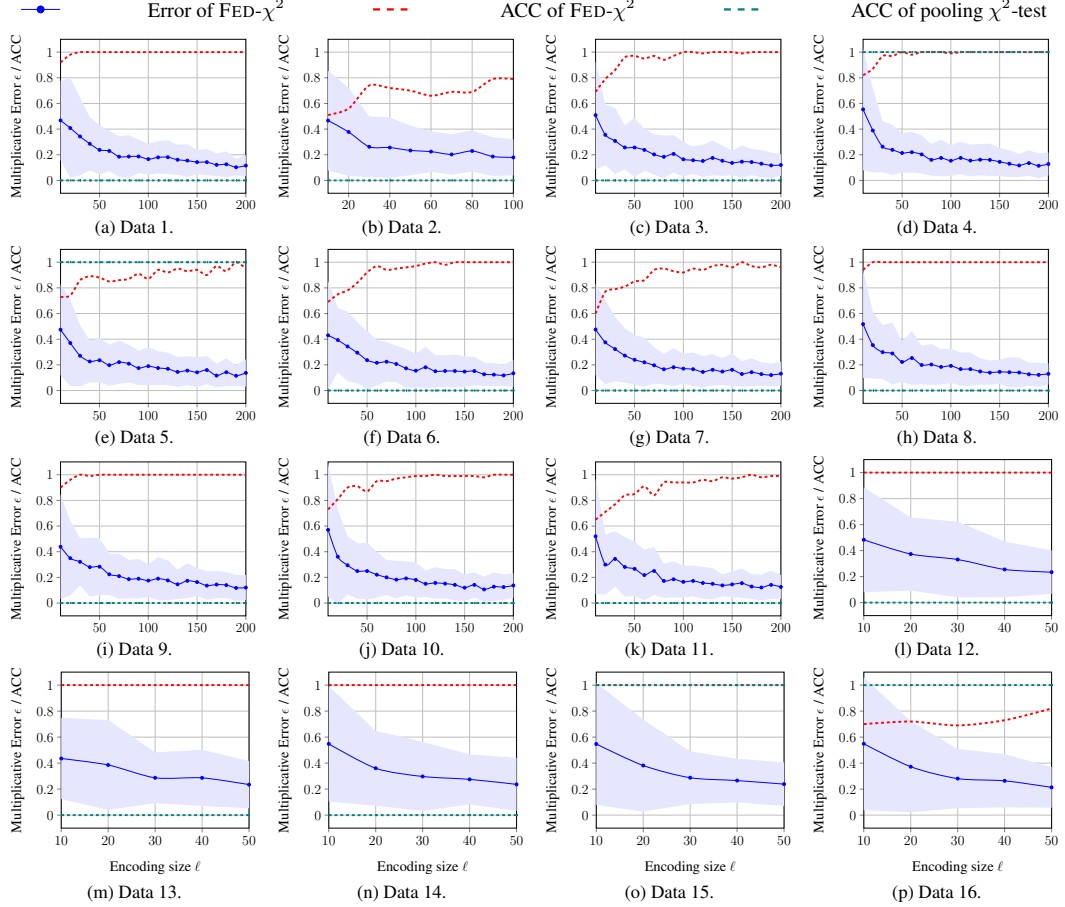

Figure 6: Comparison between FED-$\chi^2$ and pooling $\chi^2$-test.

## C  PERFORMANCE OF FED-$\chi^2$ WHEN ORIGINAL $\chi^2$-TEST ACHIEVES LOW POWER

We have computed the multiplicative error and accuracy of FED-$\chi^2$ with the original centralized $\chi^2$-test as the baseline. To further demonstrate the effectiveness of FED-$\chi^2$, we also set up the experiment to evaluate the performance of FED-$\chi^2$ on the synthesized dataset when the accuracy of the original centralized $\chi^2$-test is lower. More specifically, we synthesize the random independent, linearly correlated, quadratically correlated, and logistically correlated datasets with the same hyperparameters for 20 times. We then compute the accuracy of the original centralized $\chi^2$-test over these 20 datasets. We report the results in Fig. 7. The results show that FED-$\chi^2$ can still achieve good performance on the datasets when the original centralized $\chi^2$-test's accuracy is lower. Consistent with our results in Sec. 4.1, the multiplicative error becomes lower with the increase of the encoding size $l$. Thus, we conclude that FED-$\chi^2$'s performance is comparable to that of the original centralized $\chi^2$-test with an appropriate encoding size $l$ and this is not dependent on the datasets.

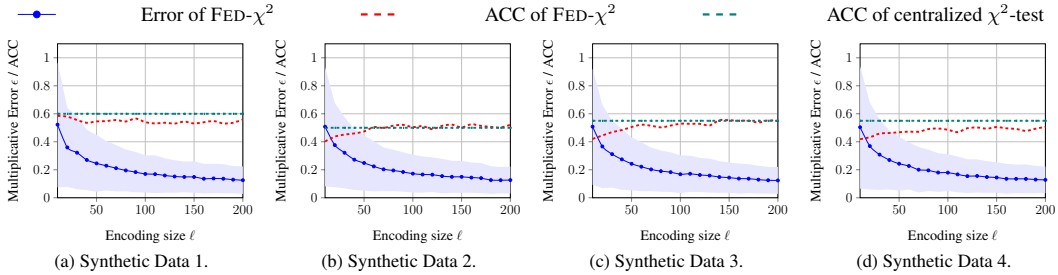

Figure 7: Performance of FED-$\chi^2$ when original $\chi^2$-test achieves low accuracy.

# D  FURTHER RESULTS FOR ONLINE FDR CONTROL

In this section, we provide further results for online FDR control. As we have shown in Fig. 5, FED-$\chi^2$ achieves good performance (FDR lower than 5%) when the encoding size $l$ is larger than 200. In Fig. 8, we provide the FDR result of the original $\chi^2$-test as well as the true discovery rate (TDR, i.e., #correct reject / #should reject). In addition, we provide statistics for each encoding size $l$ that was evaluated in Table 1. These results demonstrate that FED-$\chi^2$ performs well and is comparable to the centralized $\chi^2$-test when the encoding size $l$ is increased. More importantly, as we have shown in Sec. 4.1 and Appendix C, FED-$\chi^2$ gives results close to original centralized $\chi^2$-test and this is independent from the data.

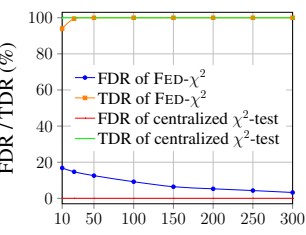

Figure 8: Results of FDR & TDR.

| | #should reject | #should accept | #correct reject | #false reject |
|---|---|---|---|---|
| FED-$\chi^2$, $l = 10$ | 5,900 | 4,100 | 5,544 | 1,132 |
| FED-$\chi^2$, $l = 25$ | 5,900 | 4,100 | 5,871 | 1,022 |
| FED-$\chi^2$, $l = 50$ | 5,900 | 4,100 | 5,899 | 856 |
| FED-$\chi^2$, $l = 100$ | 5,900 | 4,100 | 5,900 | 606 |
| FED-$\chi^2$, $l = 150$ | 5,900 | 4,100 | 5,900 | 411 |
| FED-$\chi^2$, $l = 200$ | 5,900 | 4,100 | 5,900 | 335 |
| FED-$\chi^2$, $l = 250$ | 5,900 | 4,100 | 5,900 | 270 |
| FED-$\chi^2$, $l = 300$ | 5,900 | 4,100 | 5,900 | 202 |
| centralized $\chi^2$-test | 5,900 | 4,100 | 5,900 | 0 |

Table 1: Detailed results of online FDR control.

To further demonstrate that the close performance of FED-$\chi^2$ and the original centralized $\chi^2$-test is independent of the data, we use a similar approach as described in Appendix C to generate the data that the original centralized $\chi^2$-test struggles to distinguish its correlation (i.e., resulting in a higher FDR). To achieve this, we change the covariance matrix of the Gaussian distribution when generating the independent and dependent data. The detailed results are shown in Table 2 and in Fig. 9. The original centralized $\chi^2$-test achieves an FDR of 7.03% and a TDR of 89.73%. The results indicate that when the encoding size $l$ is increased, the performance of FED-$\chi^2$ gets closer to the original centralized $\chi^2$-test. When $l = 300$, FED-$\chi^2$ performs similarly to the original centralized $\chi^2$-test. The results in Fig. 9 and Table 2 are consistent with those in Appendix C, indicating

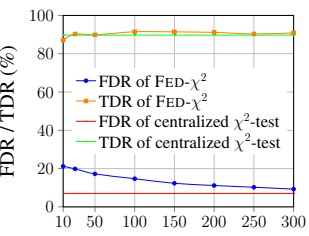

Figure 9: Results of FDR & TDR when the FDR of the original centralized $\chi^2$-test is higher.

that the performance of FED-$\chi^2$ is comparable to that of the original centralized $\chi^2$-test when the encoding size $l$ is set appropriately, and that this is independent of the data.

| | #should reject | #should accept | #correct reject | #false reject |
|---|---|---|---|---|
| FED-$\chi^2$, $l = 10$ | 5,900 | 4,100 | 5,144 | 1,392 |
| FED-$\chi^2$, $l = 25$ | 5,900 | 4,100 | 5,328 | 1,356 |
| FED-$\chi^2$, $l = 50$ | 5,900 | 4,100 | 5,325 | 1,143 |
| FED-$\chi^2$, $l = 100$ | 5,900 | 4,100 | 5,398 | 943 |
| FED-$\chi^2$, $l = 150$ | 5,900 | 4,100 | 5,393 | 765 |
| FED-$\chi^2$, $l = 200$ | 5,900 | 4,100 | 5,377 | 687 |
| FED-$\chi^2$, $l = 250$ | 5,900 | 4,100 | 5,328 | 615 |
| FED-$\chi^2$, $l = 300$ | 5,900 | 4,100 | 5,361 | 556 |
| centralized $\chi^2$-test | 5,900 | 4,100 | 5,294 | 408 |

Table 2: Detailed results of online FDR control when the FDR of the original centralized $\chi^2$-test is higher.

# E  INCORPORATE GAUSSIAN MECHANISM IN FED-$\chi^2$

As we have mentioned in Appendix A, FED-$\chi^2$ can achieve differential privacy easily by incorporating well-studied differentially private mechanisms. To further demonstrate this point, we utilize Gaussian Mechanism to provide $(\epsilon, \delta)$-DP guarantee. We clipped the local clients' data such that the encoding

function's sensitivity, which is the L2-norm of the clients' local data is bounded by $\Delta f$. Before encoding their local data, each client add Gaussian noise $\mathcal{N}^d(0, \sigma^2)$ to their local data vector $\mu_i$. We can calculate $\sigma^2 = \frac{2\ln(1.25/\delta)(\Delta f)^2}{n(\epsilon/l)^2}$. After encoding and decoding with the computed vector, FED-$\chi^2$ provides $(\epsilon, \delta)-$DP guarantee.

We evaluate the performance of differentially private FED-$\chi^2$ on the first four real-world datasets adopted in this research as shown in Fig. 10. We choose encoding size $l = 10$ and $\delta = 0.01$. The results show that the performance our algorithm becomes better with the increase of privacy budget $\epsilon$. However, when the privacy budget $\epsilon$ is small, the protocol tends to give the judgement that the data is independent because that the independent noise is too large and it dominates the test. When the privacy budget $\epsilon$ is large enough (i.e., $\epsilon = 100$), our protocol can achieve 0.92, 0.68, 0.70, and 0.80 accuracy on these datasets accordingly, and its performance is comparable to the original FED-$\chi^2$.

Again, our work is orthogonal to differential privacy, thus we leave it as future work to further study saving privacy budget, and boosting the algorithm's performance on lower $\epsilon$.

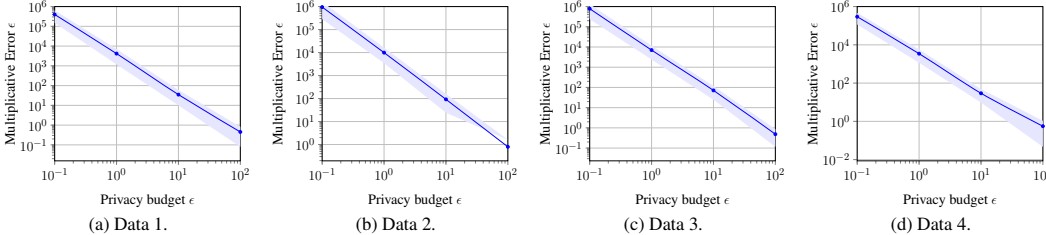

Figure 10: Performance of FED-$\chi^2$ when incorporating Gaussian Mechanism.

## F  FURTHER RESULTS FOR FEATURE SELECTION

Our results in Sec. 4.2, paragraph **Feature Selection**, demonstrate that FED-$\chi^2$ performs well when encoding size $l = 50$. We conduct experiments with different encoding sizes $l$ to further assess their effect on FED-$\chi^2$'s performance. In Fig. 11, we present the effect of encoding size $l$ on the ratio of the commonly-selected features between the original centralized $\chi^2$-test and FED-$\chi^2$. A larger ratio of commonly-selected features means that FED-$\chi^2$ performs more closely to the original centralized $\chi^2$-test. And if the ratio is 1, these two algorithms select the identical features. The results in Fig. 11 show that when the encoding size $l$ increases, the performance of FED-$\chi^2$ approaches that of the original centralized $\chi^2$-test.

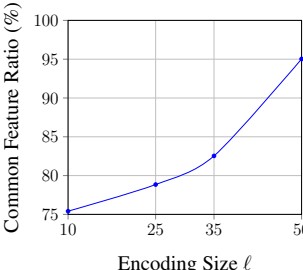

Figure 11: Ratio of commonly-selected features between FED-$\chi^2$ and original centralized $\chi^2$-test.

Similar to Sec. 4.2, we evaluate FED-$\chi^2$'s performance by training the model with the features selected by FED-$\chi^2$. Fig. 12 shows the results. When trained with FED-$\chi^2$-selected features, the model can achieve comparable accuracy to the model trained with features selected by the original centralized $\chi^2$-test. Also, consistent with the results in Fig. 3 in Sec. 4.2, we see that when the encoding size $l \geq 25$, models trained by FED-$\chi^2$-selected features achieve higher accuracy than that of the models without feature selection. These results further demonstrate the effectiveness of FED-$\chi^2$.

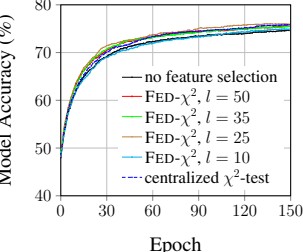

Figure 12: Accuracy of model trained w/ FED-$\chi^2$-select features under different encoding size $l$.

## G  FURTHER RESULTS ON FED-$\chi^2$ WITH DROPOUTS

We present the results of 10%, 15%, and 20% clients dropout in Fig. 13. The results further show that FED-$\chi^2$ can tolerate a considerable portion of clients dropout in Round 2 of Alg. 2.

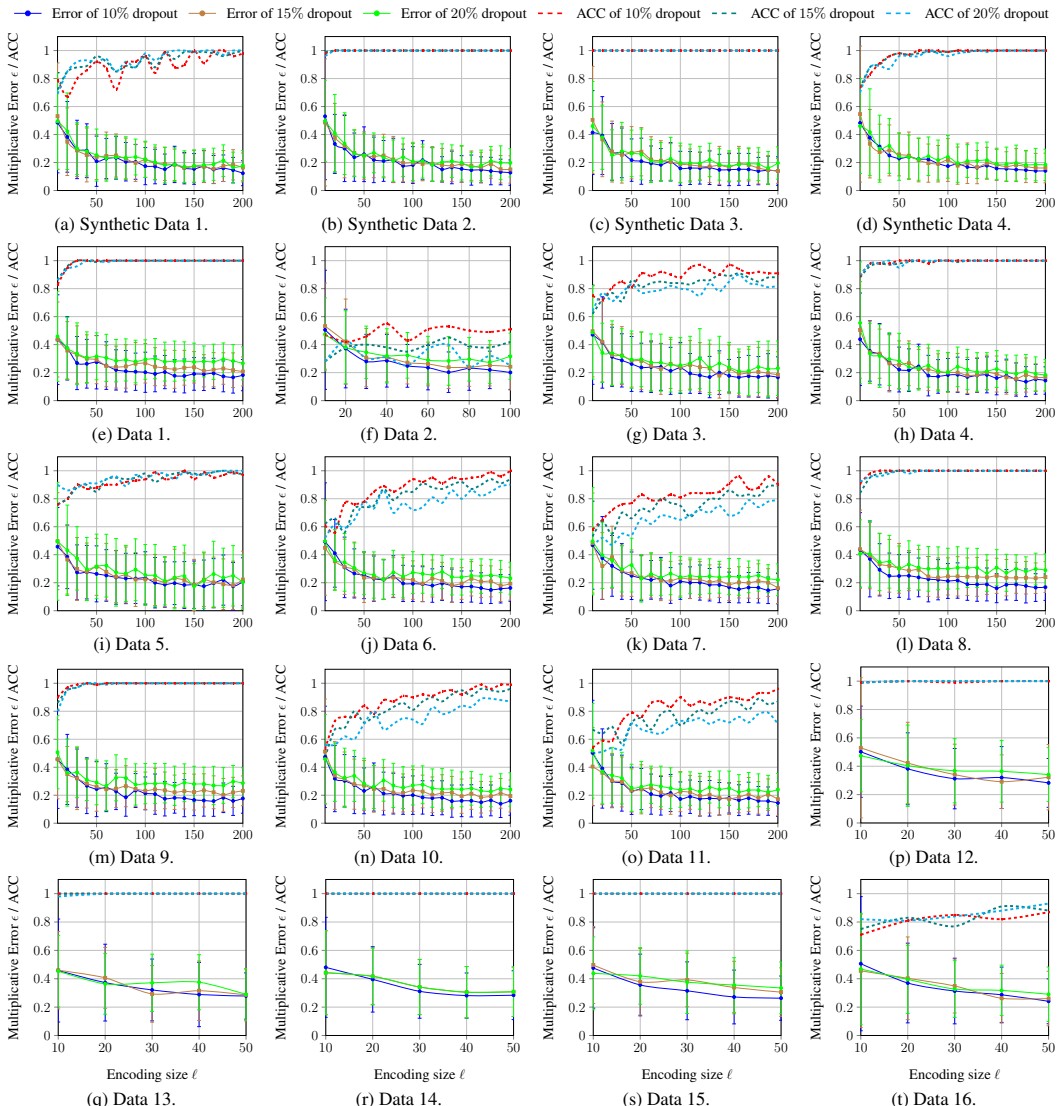

Figure 13: Multiplicative error and accuracy of FED-$\chi^2$ w.r.t. encoding size $\ell$ w/ and w/o dropout.

## H   SECURE AGGREGATION

The secure aggregation protocol from Bell et al. (2020) is presented in Alg. 4. The first step of the protocol is to generate a $k$-regular graph $G$, where the $n$ vertices are the clients participating in the protocol. The server runs a randomized graph generation algorithm INITSECUREAGG presented in Alg. 3 that takes the number of clients $n$ and samples output $(G, t, k)$ from a distribution $\mathcal{D}$. In Alg. 3, we uniformly rename the nodes of a graph known as a Harary graph defined in Definition 3 with $n$ nodes and $k$ degrees. The graph $G$ is constructed by sampling $k$ neighbours uniformly and without replacement from the set of remaining $n-1$ clients. We choose $k = \mathcal{O}(log(n))$, which is large enough to hide the updates inside the masks. $t$ is the threshold of the Shamir's Secret Sharing.

In the second step, the edges of the graph determine pairs of clients, each of which runs key agreement protocols to share random keys. The random keys will be used by each party to derive a mask for her input and enable dropouts.

In the third step, each client $c_i, i \in A_1$ sends secret share to its neighbors. In the fourth step, the server checks whether the clients dropout exceeds the threshold $\delta$, and lets the clients know their neighbors who didn't dropout.

In the fifth step, each pair $(i, j)$ of connected clients in $G$ runs a $\lambda$-secure key agreement protocol $s_{i,j} = \mathcal{KA}.Agree(sk_i^1, pk_j^1)$ which uses the key exchange in the previous step to derive a shared random key $s_{i,j}$. The pairwise masks $\mathbf{m}_{i,j} = F(s_{i,j})$ can be computed, where $F$ is the pseudorandom generator (PRG). If the semi-honest server announces dropouts and later some masked inputs of the claimed dropouts arrive, the server can recover the inputs. To prevent this happening, another level of masks, called self masks, $\mathbf{r}_i$ is added to the input. Thus, the input of client $c_i$ is: $\mathbf{y}_i = \mathbf{e}_i + \mathbf{r}_i - \sum_{j \in N_G(i), j < i} \mathbf{m}_{i,j} + \sum_{j \in N_G(i), j > i} \mathbf{m}_{i,j}$.

Steps 6–8 deal with the clients dropout by recovering the self masks $\mathbf{r}_i$ of clients who are still active and pairwise masks $\mathbf{m}_{i,j}$ of the clients who have dropped out. Finally, the server can cancel out the pairwise masks and subtract the self masks in the final sum: $\sum_{i \in A_2'} (\mathbf{y}_i - \mathbf{r}_i + \sum_{j \in NG(i) \cap (A_1' \setminus A_2'), 0 < j < i} \mathbf{m}_{i,j} - \sum_{j \in NG(i) \cap (A_1' \setminus A_2'), i < j \leq n} \mathbf{m}_{i,j})$.

**Definition 3** (HARARY$(n, k)$ Graph). *Let* HARARY$(n, k)$ *denotes a graph with $n$ nodes and degree $k$. This graph has vertices $V = [n]$ and an edge between two distinct vertices $i$ and $j$ if and only if $j - i \pmod{n} \leq (k+1)/2$ or $j - i \pmod{n} \geq n - k/2$.*

---

**Algorithm 3:** INITSECUREAGG: Generate Initial Graph for SECUREAGG.

---

1   **Function** INITSECUREAGG$(n)$:
2     $\triangleright$ $n$: Number of nodes.
3     $\triangleright$ $t$: Threshold of Shamir's Secret Sharing.
4     $k = \mathcal{O}(log(n))$.
5     Let $H =$ HARARY$(n, k)$.
6     Sample a random permutation $\pi : [n] \to [n]$.
7     Let $G$ be the set of edges $\{(\pi(i), \pi(j)) | (i, j) \in H\}$.
8     **return** $(G, t, k)$

---

## I   PROOF FOR COMMUNICATION & COMPUTATION COST

We provide the proof for Theorem 4 and Theorem 5 in the following.

**Theorem** 4 (Communication Cost). Let $\Pi$ be an instantiation of Alg. 2 with secure aggregation protocol from Bell et al. (2020), then (1) the client-side communication cost is $\mathcal{O}(\log n + m_x + m_y + \ell)$; (2) the server-side communication cost $\mathcal{O}(n \log n + nm_x + nm_y + n\ell)$.

*Proof sketch for Theorem 4.* Each client performs $k$ key agreements ($\mathcal{O}(k)$ messages, line 9 in Alg. 4) and sends 3 masked inputs ($\mathcal{O}(m_x + m_y + \ell)$ complexity, lines 3, 4, 15 in Alg. 2 and line 10 in Alg. 4). Thus, the client communication cost is $\mathcal{O}(\log n + m_x + m_y + \ell)$.

The server receives or sends $\mathcal{O}(\log n + m_x + m_y + \ell)$ messages to each client, so the server communication cost is $\mathcal{O}(n \log n + nm_x + nm_y + n\ell)$.    $\square$

**Theorem** 5 (Computation Cost). Let $\Pi$ be an instantiation of Alg. 2 with secure aggregation protocol from Bell et al. (2020), then (1) the client-side computation cost is $\mathcal{O}(m_x \log n + m_y \log n + \ell \log n + m\ell)$; (2) the server-side computation cost is $\mathcal{O}(m_x + m_y + \ell)$.

*Proof sketch for Theorem 5.* Each client computation can be broken up as $k$ key agreements ($\mathcal{O}(k)$ complexity, line 9 in Alg. 4), generating masks $\mathbf{m}_{i,j}$ for all neighbors $c_j$ ($\mathcal{O}(k(m_x + m_y + \ell))$ complexity, lines 3, 4, 15 in Alg. 2 and line 10 in Alg. 4), and encoding computation cost $\mathcal{O}(m\ell)$ (line 14 in Alg. 2). Thus, the client computation cost is $\mathcal{O}(m_x \log n + m_y \log n + \ell \log n + m\ell)$.

The server-side follows directly from the semi-honest computation analysis in Bell et al. (2020). The extra $\mathcal{O}(\ell)$ term is the complexity of the geometric mean estimator.    $\square$

---

**Algorithm 4:** SECUREAGG: Secure Aggregation Protocol. (Algorithm 2 from Bell et al. (2020))

---

1 **Function** SECUREAGG $(\{e_i\}_{i \in [n]})$ :

2    ▷ Parties: Clients $c_1, \cdots, c_n$, and Server.

3    ▷ $l$: Vector length.

4    ▷ $\mathbb{X}^l$: Input domain, $\mathbf{e}_i \in \mathbb{X}^l$.

5    ▷ $F : \{0,1\}^\lambda \to \mathbb{X}^l$: PRG.

6    ▷ *We denote by $A_1, A_2, A_3$ the sets of clients that reach certain points without dropping out. Specifically $A_1$ consists of the clients who finish step (3), $A_2$ those who finish step (5), and $A_3$ those who finish step (7). For each $A_i$, $A'_i$ is the set of clients for which the server sees they have completed that step on time.*

7    (1) The server runs $(G, t, k) = $ INITSECUREAGG $(n)$, where $G$ is a regular degree-$k$ undirected graph with $n$ nodes. By $N_G(i)$ we denote the set of $k$ nodes adjacent to $c_i$ (its neighbors).

8    (2) Client $c_i, i \in [n]$, generates key pairs $(sk_i^1, pk_i^1)$, $(sk_i^2, pk_i^2)$ and sends $(pk_i^1, pk_i^2)$ to the server who forwards the message to $N_G(i)$.

9    (3) **for** *each Client $c_i, i \in A_1$* **do**

       • Generates a random PRG seed $b_i$.

       • Computes two sets of shares:

$$H_i^b = \{h_{i,1}^b, \cdots, h_{i,k}^b\} = ShamirSS(t, k, b_i)$$

$$H_i^s = \{h_{i,1}^s, \cdots, h_{i,k}^s\} = ShamirSS(t, k, sk_i^1)$$

       • Sends to the server a message $m = (j, c_{i,j})$, where $c_{i,j} = \mathcal{E}_{auth}.Enc(k_{i,j}, (i||j||h_{i,j}^b||h_{i,j}^s))$ and $k_{i,j} = \mathcal{KA}.Agree(sk_i^2, pk_j^2)$, for each $j \in N_G(i)$.

10    (4) The server aborts if $|A'_1| < (1 - \delta)n$ and otherwise forwards $(j, c_{i,j})$ to client $c_j$ who deduces $A'_1 \cap N_G(j)$.

11    (5) **for** *each Client $c_i, i \in A_2$* **do**

       • Computes a shared random PRG seed $s_{i,j}$ as $s_{i,j} = \mathcal{KA}.Agree(sk_i^1, pk_j^1)$.

       • Computes masks $\mathbf{m}_{i,j} = F(s_{i,j})$ and $\mathbf{r}_i = F(b_i)$.

       • Sends to the server their masked input

$$\mathbf{y}_i = \mathbf{e}_i + \mathbf{r}_i - \sum_{j \in [n], j < i} \mathbf{m}_{i,j} + \sum_{j \in [n], j > i} \mathbf{m}_{i,j}$$

12    (6) The server collects masked inputs. It aborts if $|A'_2| < (1 - \delta)n$ and otherwise sends $(A'_2 \cup N_G(i), (A_1 \backslash A'_2) \cup N_G(i))$ to every client $c_i, i \in A'_2$.

13    (7) Client $c_j, j \in A_3$ receives $(R_1, R_2)$ from the server and sends $\{(i, h_{i,j}^b)\}_{i \in R_1} \cup \{(i, h_{i,j}^s)\}_{i \in R_2}$ obtained by decrypting the $c_{i,j}$ received in Step (3).

14    (8) The server aborts if $|A'_3| < (1 - \delta)n$ and otherwise:

       • Collects, for each client $c_i, i \in A'_2$, the set $B_i$ of all shares in $H_i^b$ sent by clients in $A_3$. Then aborts if $|B_i| < t$ and otherwise recovers $b_i$ and $\mathbf{r}_i$ using the $t$ shares received which came from the lowest client IDs.

       • Collects, for each client $c_i, i \in (A_1 \backslash A'_2)$, the set $S_i$ of all shares in $H_i^s$ sent by clients in $A_3$. Then aborts if $|S_i| < t$ and otherwise recovers $sk_i^1$ and $\mathbf{m}_{i,j}$.

       • **return** $\sum_{i \in A'_2}(\mathbf{y}_i - \mathbf{r}_i + \sum_{j \in NG(i) \cap (A'_1 \backslash A'_2), 0 < j < i} \mathbf{m}_{i,j} - \sum_{j \in NG(i) \cap (A'_1 \backslash A'_2), i < j \leq n} \mathbf{m}_{i,j})$.

---

## J DETAILS OF DATASETS

The details for the real-world datasets used in Sec. 4.1 are provided in Table 3. The license of Credit Risk Classification (Govindaraj, Praveen) is CC BY-SA 4.0, the license of German Traffic

Sign (Houben et al., 2013) is CC0: Public Domain. Other datasets without a license are from UCI Machine Learning Repository (Dua & Graff, 2017).

Table 3: Dataset details.

| ID | Data | Attr #1 | A#1 Cat | Attr #2 | A#2 Cat |
|---|---|---|---|---|---|
| 1 | Adult Income (Kohavi, 1996; Kohavi, Ronny and Becker, Barry) | Occupation | 14 | Native Country | 41 |
| 2 | Credit Risk Classification (Govindaraj, Praveen) | Feature 6 | 14 | Feature 7 | 11 |
| 3 | Credit Risk Classification (Govindaraj, Praveen) | Credit Product Type | 28 | Overdue Type I | 35 |
| 4 | Credit Risk Classification (Govindaraj, Praveen) | Credit Product Type | 28 | Overdue Type II | 35 |
| 5 | Credit Risk Classification (Govindaraj, Praveen) | Credit Product Type | 28 | Overdue Type III | 36 |
| 6 | German Traffic Sign (Houben et al., 2013) | Image Width | 219 | Traffic Sign | 43 |
| 7 | German Traffic Sign (Houben et al., 2013) | Image Height | 201 | Traffic Sign | 43 |
| 8 | German Traffic Sign (Houben et al., 2013) | Upper left X coordinate | 21 | Traffic Sign | 43 |
| 9 | German Traffic Sign (Houben et al., 2013) | Upper left Y coordinate | 16 | Traffic Sign | 43 |
| 10 | German Traffic Sign (Houben et al., 2013) | Lower right X coordinate | 204 | Traffic Sign | 43 |
| 11 | German Traffic Sign (Houben et al., 2013) | Lower right Y coordinate | 186 | Traffic Sign | 43 |
| 12 | Mushroom (Schlimmer, Jeff) | Cap color | 10 | Odor | 9 |
| 13 | Mushroom (Schlimmer, Jeff) | Gill color | 12 | Stalk color above ring | 9 |
| 14 | Mushroom (Schlimmer, Jeff) | Stalk color below ring | 9 | Ring Type | 8 |
| 15 | Mushroom (Schlimmer, Jeff) | Spore print color | 9 | Habitat | 7 |
| 16 | Lymphography (Kononenko, Igor and Cestnik, Bojan) | Structure Change | 8 | No. of nodes | 8 |

## K  DETAILS OF REGRESSION MODELS

The details of the regression models trained in feature selection in Sec. 4.2 is reported in Table 4. The training and testing splits are the same for FED-$\chi^2$, centralized $\chi^2$-test and model without feature selection (i.e. there are 17,262 training and 4,316 test documents). We use the same learning rate; random seed and all other settings are also the same to make the comparison fair. We get the result of Fig. 3 and the models are all trained on NVIDIA GeForce RTX 3090.

Table 4: Model details.

| Task | Model Size | Learning Rate | Random Seed |
|---|---|---|---|
| FED-$\chi^2$ | $40000 \times 20$ | 0.1 | 0 |
| Centralized $\chi^2$-test | $40000 \times 20$ | 0.1 | 0 |
| Without Feature Selection | $167135 \times 20$ | 0.1 | 0 |

## L  SAFFRON PROCEDURE

In Sec. 4.2, we adopt the SAFFRON procedure (Ramdas et al., 2018) to perform online FDR control. SAFFRON procedure is currently the state of the arts for multiple hypothesis testing. In Alg. 5, we formally present the SAFFRON algorithm.

The initial error budget for SAFFRON is $(1 - \lambda_1 W_0) < (1 - \lambda_1 \alpha)$, and this will be allocated to different tests over time. The sequence $\{\lambda_j\}_{j=1}^{\infty}$ is defined by $g_t$ and $\lambda_j$ serves as a weak estimation of $\alpha_j$. $g_t$ can be any coordinate wise non-decreasing function (line 8 in Alg. 5). $R_j := I(p_j < \alpha_j)$ is the indicator for rejection, while $C_j := I(p_j < \lambda_j)$ is the indicator for candidacy. $\tau_j$ is the $j^{th}$ rejection time. For each $p_t$, if $p_t < \lambda_t$, SAFFRON adds it to the candidate set $C_t$ and sets the candidates after the $j^{th}$ rejection (lines 9-10 in Alg. 5). Further, the $\alpha_t$ is updated by several parameters like current wealth, current total rejection numbers, the current size of the candidate set, and so on (lines 11-14 in Alg. 5). Then, the decision $R_t$ is made according to the updated $\alpha_t$ (line 15 in Alg. 5).

The hyper-parameters we use for the SAFFRON procedure in online false discovery rate control of Sec. 4 are aligned with the setting in Ramdas et al. (2018). In particular, the target FDR level is $\alpha = 0.05$, the initial wealth is $W_0 = 0.0125$, and $\gamma_j$ is calculated in the following way: $\gamma_j = \frac{1/(j+1)^{1.6}}{\sum_{j=0}^{10000} 1/(j+1)^{1.6}}$.

**Algorithm 5:** SAFFRON Procedure.

**1 Function** SAFFRONPROCEDURE ($\{p_1, p_2, \cdots\}$, $\alpha$, $W_0$, $\{\gamma_j\}_{j=0}^{\infty}$)**:**

**2** $\quad \triangleright \{p_1, p_2, \cdots\}$: Stream of $p$-values.

**3** $\quad \triangleright \alpha$: Target FDR level.

**4** $\quad \triangleright W_0$: Initial wealth.

**5** $\quad \triangleright \{\gamma_j\}_{j=0}^{\infty}$: Positive non-increasing sequence summing to one.

**6** $\quad i \leftarrow 0$                  `// Set rejection number.`

**7** $\quad$ **for** *each p-value $p_t \in \{p_1, p_2, \cdots\}$* **do**

**8** $\quad\quad \lambda_t \leftarrow g_t(R_{1:t-1}, C_{1:t-1})$

**9** $\quad\quad C_t \leftarrow I(p_t < \lambda_t)$         `// Set the indicator for candidacy` $C_t$`.`

**10** $\quad\quad C_{j+} \leftarrow \sum_{i=\tau_j+1}^{t-1} C_i$       `// Set the candidates after the` $j^{th}$

$\quad\quad$ `rejection.`

**11** $\quad\quad$ **if** $t = 1$ **then**

**12** $\quad\quad\quad \alpha_1 \leftarrow (1 - \lambda_1)\gamma_1 W_0$

**13** $\quad\quad$ **else**

**14** $\quad\quad\quad \alpha_t \leftarrow (1 - \lambda_t)(W_0 \gamma_{t-C_{0+}} + (\alpha - W_0)\gamma_{t-\tau_1-C_{1+}} + \sum_{j\geq 2} \alpha \gamma_{t-\tau_j-C_{j+}})$

**15** $\quad\quad R_t \leftarrow I(p_t \leq \alpha_t)$                `// Output` $R_t$`.`

**16** $\quad\quad$ **if** $R_t = 1$ **then**

**17** $\quad\quad\quad i \leftarrow i + 1$          `// Update rejection number.`

**18** $\quad\quad\quad \tau_i \leftarrow t$             `// Set the` $i^{th}$ `rejection time.`

**19** $\quad$ **return** $\{R_0, R_1, \cdots\}$

