# OpenReview forum: "FED-$\chi^2$: Secure Federated Correlation Test"
_ICLR.cc/2022/Conference — ICLR 2022 Submitted_

### Official Review · Reviewer_GEWR · 2021-10-29

**Correctness:** 2
**Technical Novelty And Significance:** 3
**Empirical Novelty And Significance:** 3
**Recommendation:** 5
**Confidence:** 2

**Main Review:**

Strengths:

1) The problem is important and complex.

2) The numeric experimental results are somewhat convincing.

Weakness:

1) I could not find any plausible proof or development of the theoretical claims. This may be a matter of presentation, but Sections~3.5-3.7 seem to be a bunch of statements without justification, explanation, mutual linkages or proofs.

2) The paper is hard to follow and seems hastily written without proper development. For example, Section~3.2 seems to end abruptly. It describes a challenge, but not how the paper's proposal addresses the challenge. Such details may be elsewhere, but is unclear from the narrative.

3) Not considering differential privacy seems a major deficiency.




**Summary Of The Paper:**

The paper proposed a federated $\chi^{2}$-test protocol. This is proposed as a technique that is computation- and communication-efficient and leaks much less information than other alternatives. The claim is that the proposed technique can tolerate up to 20\% of clients dropout with minor accuracy drop. The proposed technique does not include differential privacy guarantee.


**Summary Of The Review:**

The problem is important. However, technical concerns remain and the paper does not seem to be well-developed and is hard to read.


Revised comments: Upon reading the authors' comments, I understand their perspective on proofs (not one that I completely agree with) and I thank them for pointing out the places where proofs may be found. However, I am still not convinced that the writing is transparent and clear, or that the issue of differential privacy may be entirely waved off. That being said, I will revise my rating and downgrade my confidence/expertise score, so if the paper is appealing to other reviewers, I will not stand in the way of its acceptance.

---

> ### Author Response · Authors · 2021-11-14
> **Response to Reviewer GEWR**
>
> We would like to thank reviewer GEWR for the detailed review. There seem to be several major confusions caused by our lack of elaboration. We hope this response can help clarify them. We also kindly ask the reviewer to correspondingly increase the score as some of these confusions are major and might influence the reviewer’s initial judgement.
>
> ***
>
> **Q1:** I could not find any plausible proof or development of the theoretical claims. This may be a matter of presentation, but Sections~3.5-3.7 seem to be a bunch of statements without justification, explanation, mutual linkages or proofs.
>
> **A1:** We politely disagree with the reviewer because each Theorem in the paper is supported by a proof. Theorem 1 and 2’s proofs are inline in Section 3.5. One possible confusion here is the proof of Theorem 1 is a simulation-based proof [1] which might seem unfamiliar to non-experts in cryptography. Theorem 3’s proof is inline in Section 3.6. Theorem 4 and 5’s proofs are deferred to Appendix D (Appendix I in our newest version) due to the space limit. Also, we have cited [1] to help the audience without crypto background better understand simulation-based proof. Please let us know if there is any proof that we have missed. If this addresses the reviewer’s concern, we kindly ask for an increase in the score accordingly as we assume this confusion might have greatly influenced the reviewer’s judgement initially.
>
> ***
>
> **Q2:** The paper is hard to follow and seems hastily written without proper development. For example, Section~3.2 seems to end abruptly. It describes a challenge, but not how the paper's proposal addresses the challenge. Such details may be elsewhere, but is unclear from the narrative.
>
> **A2:** We apologize for the confusion caused by our lack of elaboration. The main purpose of Section 3.2 is to recast $\chi^2$-test to second frequency moment estimation, which is then solved throughout the rest of Section 3. Section 3.2 might be short but it is logically independent from the other sections so we make it an independent subsection.
>
> We agree that our paper might not be perfect and have issues in presentation but we do take several passes through it to make sure it is in good enough shape to submit. The other reviewers do not criticize the writing either so we politely disagree with the reviewer that the paper is hastily written, which we take as serious criticism on the writing quality of the paper.
>
> ***
>
> **Q3:** Not considering differential privacy seems a major deficiency.
>
> **A3:** Thanks for the feedback! Secure aggregation is a security guarantee orthogonal to differential privacy so we do not consider DP in the present work as they are not technically correlated as stated in Section 2 and Appendix A. As a result, we politely disagree that it is a major deficiency. On the other hand, DP can be easily added using standard mechanisms such as discrete Gaussian mechanism [2]. We are currently running experiments with DP and will append to the response once it is ready.
>
> [1] Lindell, Yehuda. "How to simulate it–a tutorial on the simulation proof technique." Tutorials on the Foundations of Cryptography (2017): 277-346.
> [2] Kairouz, Peter, Ziyu Liu, and Thomas Steinke. "The distributed discrete gaussian mechanism for federated learning with secure aggregation." arXiv preprint arXiv:2102.06387 (2021).

---

> ### Author Response · Authors · 2021-11-22
> **A kind reminder before the discussion phase ends**
>
> As the discussion phase is ending tomorrow, we would like to thank you again for your constructive reviews. We would appreciate it if you can let us know whether our rebuttal has addressed any of your concerns and whether you have any follow-up questions.

---

### Official Review · Reviewer_sPzq · 2021-11-01

**Correctness:** 3
**Technical Novelty And Significance:** 3
**Empirical Novelty And Significance:** Not applicable
**Recommendation:** 6
**Confidence:** 2

**Main Review:**

The algorithm presented in the paper is pretty simple. First, it uses secure aggregation to compute marginal statistics. Each participant applies a stable random projection (with particular parameters) on their (normalized) frequency distribution. The results are then aggregated again using secure aggregation. Due to the nature of the stable distribution, the aggregated result follows a stable distribution as well, and its scale can be estimated. This scale ends up being equal to the squared sum in the correlation coefficient.

One concern I have is that the protocol actually reveals some significant amount of information to the server in addition to just the result. The authors explain in a remark that this is not a concern, because the leaked information constitutes a linear system (over a finite field) with a large solution space. However, I don't understand if all of these solutions are supposed to be equally realistic or equally likely. For example, realistic solutions may be particularly short vectors, or have some other properties that may make identifying them possible. Does this make it easier to find a solution to the system?

**Summary Of The Paper:**

This paper introduces a federated analytics technique for computing the Pearson correlation for two random variables. The method is based on repeated uses of secure aggregation, as well as stable random projections. The protocol is argued to be secure in a semi-honest security model.

**Summary Of The Review:**

This paper presents a nice new federated analytics technique for computing correlations. I have some concerns about the security of the protocol, but it's possible I'm misunderstanding something.

---

> ### Author Response · Authors · 2021-11-17
> **Response to Reviewer sPzq**
>
> We would like to thank Reviewer sPzq for the insightful comments. We reply to the questions inline as below.
>
> ***
>
> Q1: One concern I have is that the protocol actually reveals some significant amount of information to the server in addition to just the result. The authors explain in a remark that this is not a concern, because the leaked information constitutes a linear system (over a finite field) with a large solution space. However, I don't understand if all of these solutions are supposed to be equally realistic or equally likely. For example, realistic solutions may be particularly short vectors, or have some other properties that may make identifying them possible. Does this make it easier to find a solution to the system?
>
> A1: Thanks for the insightful comment! We agree with you that it becomes easier to find a solution to the system. Overall, auxiliary information can assist in further reducing the number of impossible solutions over the entire solution space. However, we wish to clarify that this issue is not unique to our work; it is a problem that affects all secure aggregation techniques, to the best of our knowledge. On the other hand, our technique leaks significantly less information than collecting the joint distribution directly, ensuring that the security guarantee is rigorously and significantly better than the trivial solution. This will be clarified in revision.
>
> ***

---

> ### Author Response · Authors · 2021-11-22
> **A kind reminder before the discussion phase ends**
>
> As the discussion phase is ending tomorrow, we would like to thank you again for your constructive reviews. We would appreciate it if you can let us know whether our rebuttal has addressed any of your concerns and whether you have any follow-up questions.

---

### Official Review · Reviewer_DQ1G · 2021-11-02

**Correctness:** 3
**Technical Novelty And Significance:** 2
**Empirical Novelty And Significance:** 2
**Recommendation:** 5
**Confidence:** 2

**Main Review:**

Strengths:
1. The algorithm is intuitive and easy to implement, and several possible applications are discussed.
2. It provides the relationship of the reduced dimension with the test accuracy and the computation cost.

Weaknesses:
1. In evaluation section 4.1, what is the accuracy of the original chi-square test? If the original test has power 1 in the presented simulation, would we expect a similar comparison with the proposed test when the original test has lower power, such as 0.5?
2. I don’t fully understand Figure 3. What is an epoch? When computing the accuracy, what is the ground truth?
3. In the case study of online FDR, could you also show the power (#correctly rejected/ #hypotheses should be rejected)? Although unlikely, I would want to know whether lower FDR as $l$ increase is because the test is conservative and the power is also lower. Could you also include the original chi-square test in this case?
4. Theoretically, do we know if the FED chi-square test statistics is an unbiased estimation given the original chi-square statistics, or systematically larger or smaller?
5. As a reader not in the expertise of stable random projection and geometric mean estimator, I don't fully understand why the lower dimensional $e_k$ gets to the approximation.


**Summary Of The Paper:**

This paper discusses the problem of conducting correlation tests with sensitive data separately collected from $n$ clients, where centralizing data collection is risky but the available secure multiparty computation is costly in computation. The proposed test adjusts the classical chi-square test by first rewriting it as a second frequency moments estimation using $n$ vectors, each from a client. The proposed test prevent data recovering when performing the test by letting each client project its vector to a lower dimension. The higher the dimension is, the more accurate the test is, yet the higher the computation costs.


**Summary Of The Review:**

Overall I think the proposed test is intuitive, but I am a little confused about the experiments and theory as described above. I would feel more confident to judge the paper if my confusion is resolved.

---

> ### Author Response · Authors · 2021-11-17
> **Response to Reviewer DQ1G**
>
> We would like to thank Reviewer DQ1G for the insightful comments. We reply to the questions inline as below.
>
> ***
>
> Q1: In evaluation section 4.1, what is the accuracy of the original chi-square test? If the original test has power 1 in the presented simulation, would we expect a similar comparison with the proposed test when the original test has lower power, such as 0.5?
>
> A1: Thanks for the insightful comment! As suggested, we set up an experiment to evaluate the performance of Fed-$\chi^2$ on the synthesized datasets on which the accuracy of the centralized $\chi^2$-test is low. We report the results in **Appendix C**. The results show that Fed-$\chi^2$ can achieve comparable performance to the centralized $\chi^2$-test on these datasets with proper encoding size.
>
> ***
>
> Q2: I don’t fully understand Figure 3. What is an epoch? When computing the accuracy, what is the ground truth?
>
> A2: Epoch here is the number of passes that LR runs on the training dataset with only selected features. Accuracy is the LR’s accuracy on a test dataset. We evaluate the performance of Fed-$\chi^2$ in feature selection by the accuracy of the model trained on the selected features.
>
> ***
>
> Q3: In the case study of online FDR, could you also show the power (#correctly rejected/ #hypotheses should be rejected)? Although unlikely, I would want to know whether lower FDR as $l$ increase is because the test is conservative and the power is also lower. Could you also include the original chi-square test in this case?
>
> A3: Thanks for the comments! We provide further results of online FDR control in **Appendix D**. The lower FDR as $l$ increase is because the estimation error of Fed-$\chi^2$ is smaller and its result is closer to the centralized $\chi^2$-test, which is consistent with our conclusion in Section 4.1.
>
> ***
>
> Q4: Theoretically, do we know if the FED chi-square test statistics is an unbiased estimation given the original chi-square statistics, or systematically larger or smaller?
>
> A4: The result of Fed-$\chi^2$ is unbiased, since all steps in our protocol, including encoding with stable projection, aggregation with Secure Aggregation, and decoding with Geometric Mean Estimator are unbiased.
>
> ***
>
> Q5: I don't fully understand why the lower dimensional $e_k$ gets to the approximation.
>
> A5: $e_k$ can be viewed as a set of random variables whose distribution scale is a function of the second frequency moments. Thus, by estimating the scale of the distribution using $e_k$, we can get the approximation.
>
> ***

---

> > ### Comment · Reviewer_DQ1G · 2021-11-20
> > **Further questions on Q2 and Q3**
> >
> > Q2: Thank you for the explanation. I feel perhaps varying the encoding size $l$ would be more informative. Is it computationally intensive?
> >
> > Q3: Are you saying the procedure using the standard chi-square test has FDR zero? Does it mean the simulated situation is too optimal (easy for identification)? Can you describe what's the null and alternative hypothesis, and how is the data generated for each hypothesis? I understand it's a multivariate Gaussian, but are you testing the mean, and what is the true mean?

---

> > > ### Author Response · Authors · 2021-11-21
> > > **Response to further questions on Q2 and Q3**
> > >
> > > Thanks for the constructive comments. If the below answers address your concern, we would like to politely ask for an increase in score. Please let us know if you have any further concerns or suggestions, and we will gladly respond and extend our work.
> > >
> > > ***
> > >
> > > Q2: Thank you for the explanation. I feel perhaps varying the encoding size  would be more informative. Is it computationally intensive?
> > >
> > > A2: In **Appendix F** of our paper, we examined Fed-$\chi^2$’s performance on feature selection under different encoding sizes $l$. The results in Fig.11 and Fig.12 demonstrate that when the encoding size $l$ increases, Fed-$\chi^2$ achieves improved performance and the ratio of the commonly-selected features between Fed-$\chi^2$ and the original centralized $\chi^2$-test increases as well.
> > >
> > > ***
> > >
> > > Q3: Are you saying the procedure using the standard chi-square test has FDR zero? Does it mean the simulated situation is too optimal (easy for identification)? Can you describe what's the null and alternative hypothesis, and how is the data generated for each hypothesis? I understand it's a multivariate Gaussian, but are you testing the mean, and what is the true mean?
> > >
> > > A3: We clarify that Fed-$\chi^2$ performs closely to the original centralized $\chi^2$-test under proper encoding size $l$ (e.g., $l=50$ in FDR control). And this is independent from the data, as we also noted and evaluated in our previous response to Q1 in **Appendix C**.
> > >
> > > To further demonstrate this, we have set up the experiment in the second paragraph of **Appendix D** to evaluate the performance of Fed-$\chi^2$ with the data that the FDR of the original centralized $\chi^2$-test is higher (i.e., 7.03%). The results are consistent with our prior findings in Fig.8. Please refer to the second paragraph of **Appendix D** for further explanation.
> > >
> > > As for the data generation, we generate the local contingency table with Gaussian distribution. That is, we generate independent data using a random mean and the identity matrix E as its covariance matrix. We generate dependent data using a random mean and a covariance matrix whose off-diagonal elements are not always zero (the covariance of different items in the table is not 0).

---

> > > ### Author Response · Authors · 2021-11-30
> > > **A kind reminder before the end of the discussion period**
> > >
> > > Dear reviewer,
> > >
> > > As the discussion period is ending soon, we would like to know whether the added experiments have addressed your questions. If they do, we would like to politely ask for an increase in score. Please let us know if you have any further concerns or suggestions, and we will gladly respond and extend our work.
> > >
> > > Cheers,
> > >
> > > Anonymous Authors

---

### Official Review · Reviewer_UTuW · 2021-11-08

**Correctness:** 3
**Technical Novelty And Significance:** 3
**Empirical Novelty And Significance:** 3
**Recommendation:** 6
**Confidence:** 2

**Main Review:**

Paper proposes Fed-Chi square,a  secure chi-square test in the federated setting. The proposed method is practical in that it has many real-world applications as discussed by the authors. Overall, I like the paper, I have couple questions for the authors:

1. For security, although the paper states that the method does not guarantee differential privacy, is it possible to show the gain in accuracy after not considering DP?

2. Also, how does this compare to standard pooling of test statistics? i.e. everyone does the test on their own data and the final estimates are pooled. Similar to how one does for multiple imputation.

**Summary Of The Paper:**

Paper proposes chi-square test in the federated setting where data sharing is not possible.

**Summary Of The Review:**

It is a promising paper, I have couple minor questions.

---

> ### Author Response · Authors · 2021-11-17
> **Response to Reviewer UTuW**
>
> We would like to thank Reviewer UTuW for the insightful comments. We reply to the questions inline as below.
>
> ***
>
> Q1: For security, although the paper states that the method does not guarantee differential privacy, is it possible to show the gain in accuracy after not considering DP?
>
> A1: Thanks for the comment! Although our work is orthogonal to DP as stated in Sec. 3.1, it is easy to add DP to it by introducing calibrated discrete Gaussian noise to the users’ local updates [1]. We are currently running experiments with DP and will append to the response once it is ready.
>
> [1] Kairouz, Peter, Ziyu Liu, and Thomas Steinke. "The distributed discrete gaussian mechanism for federated learning with secure aggregation." arXiv preprint arXiv:2102.06387 (2021).
>
> ***
>
> Q2: How does this compare to standard pooling of test statistics? i.e. everyone does the test on their own data and the final estimates are pooled. Similar to how one does for multiple imputation.
>
> A2: Thanks for the insightful comment! As suggested, we set up an experiment to evaluate the performance of the pooling algorithm on the real-world datasets. The results are shown in Figure 6 in **Appendix B**. We can tell that simple pooling could not provide meaningful information especially when the number of each client’s observations is not sufficient. This comparison further demonstrates the effectiveness and necessity of Fed-$\chi^2$.
>
> ***

---

> ### Author Response · Authors · 2021-11-20
> **Follow-up on the DP experiments**
>
> We have finished the DP-related experiments and provide a more complete answer to the below question.
>
> ***
>
> Q: For security, although the paper states that the method does not guarantee differential privacy, is it possible to show the gain in accuracy after not considering DP?
>
> A: We have provided the evaluation results of Fed-$\chi^2$ in **Appendix E** of the paper when incorporated with Gaussian Mechanism. The results show that trivially adding Gaussian noise to achieve DP results in a drastic accuracy drop, especially when the privacy budget $\epsilon$ is small. In other words, the gain in accuracy is seen as **large** after not considering DP. Again, our work is orthogonal to differential privacy, and we leave it as future work to further study saving privacy budget and boosting the algorithm's performance on lower $\epsilon$.
>
> ***

---

> > ### Comment · Reviewer_UTuW · 2021-11-21
> > **Follow-up on the DP experiments**
> >
> > Thank you for adding the extra results. It certainly makes the paper stronger.

---

### Decision · Program_Chairs · 2022-01-20

**Decision:**

Reject

**Comment:**

This work proposes a federated version of the classical $\chi^2$ correlation test. The key new step is the use of stable projection to reduce computational overheads associated with the use of secure multi-party protocols. Overall while the contribution is of interest the novelty is rather limited. I also consider the work to be somewhat outside of scope for ICLR. It would be more suitable for a security or statistics focused venue. Therefore I do not recommend acceptance.